# Probing Dynamic Variation of Layered Microstructure Using Backscattering Polarization Imaging

**Tongjun Bu** [1,†], **Conghui Shao** [2,†], **Yuanhuan Zhu** [3], **Tongyu Huang** [4], **Qianhao Zhao** [1], **Yanan Sun** [5], **Yi Wang** [5] **and Hui Ma** [1,2,3,4,*]

1   Shenzhen International Graduate School, Tsinghua University, Shenzhen 518055, China;
btj20@mails.tsinghua.edu.cn (T.B.); zqh20@mails.tsinghua.edu.cn (Q.Z.)
2   Department of Physics, Tsinghua University, Beijing 100084, China; sch18@mails.tsinghua.edu.cn
3   Tsinghua-Berkeley Shenzhen Institute, Tsinghua University, Shenzhen 518055, China;
zhu-yh16@mails.tsinghua.edu.cn
4   Department of Biomedical Engineering, Tsinghua University, Beijing 100084, China;
hty19@mails.tsinghua.edu.cn
5   Experimental Research Center, China Academy of Chinese Medicine Science, Beijing 100091, China;
sunyanan@merc.ac.cn (Y.S.); wangyi@merc.ac.cn (Y.W.)
*   Correspondence: mahui@tsinghua.edu.cn
†   These authors contributed equally to this work.

**Abstract:** Polarization imaging can quantitatively probe the microscopic structure of biological tissues which can be complex and consist of layered structures. In this paper, we established a fast-backscattering Mueller matrix imaging system to characterize the dynamic variation in the microstructure of single-layer and double-layer tissues as glycerin solution penetrated into the samples. The characteristic response of Mueller matrix elements, as well as polarization parameters with clearer physics meanings, show that polarization imaging can capture the dynamic variation in the layered microstructure. The experimental results are confirmed by Monte Carlo simulations. Further examination on the accuracy of Mueller matrix measurements also shows that much faster speed has to be considered when backscattering Mueller matrix imaging is applied to living samples.

**Keywords:** dynamics; polarization; backscattering; Mueller matrix

## 1. Introduction

Polarization imaging has many advantages such as being non-invasive, sensitive to microstructural features [1–7], and capable of quantitative detection of microstructural features in biological tissues [8–10]. Polarization imaging can be achieved by adding a polarization states generator and analyzer to existing optical systems. For example, common optical microscopes can be upgraded for polarization imaging [11]. Compared with traditional non-polarized optical methods, a polarization measurement can provide much richer information on the microstructure of scattering samples such as biological tissues [12]. Such measurements are label-free and non-invasive and are therefore attractive tools for monitoring dynamic processes in living samples.

The states of polarization of light can be described by a four-dimensional Stokes vector $S$. The polarization property of the sample can be characterized by a $4 \times 4$ Mueller matrix [12]. As shown in Equation (1), the Mueller matrix is the transformation matrix of polarization states.

$$S_{out} = MS_{in}$$
$$M = \begin{bmatrix} M11 & M12 & M13 & M14 \\ M21 & M22 & M23 & M24 \\ M31 & M32 & M33 & M34 \\ M41 & M42 & M43 & M44 \end{bmatrix} \tag{1}$$

Stacking two layers of simple optical properties can result in Mueller matrices (MM) of much more complicated forms [13,14]. Identifying the characteristic behavior of a layered structure can be very helpful to characterize the microstructure of complex tissues.

In our previous works, we used multi-color backscattering Mueller matrix imaging (MMI) based on dual rotating retarders (DRRs) [15] to probe the layered skin of mice, since different wavelengths reach different penetration depths [16]. Using experiments on fabricated samples mimicking a layered structure and Monte Carlo simulations based on the sphere-cylinder birefringence model (SCBM), it was proved that specific polarization parameters can be used to identify layered tissues and extract quantitative information on the layers.

Tissue optical clearing (TOC) is a convenient technique to increase the penetration depth of thick tissues [17–22]. The replacement of free water with optical clearing agents of higher refractive index, such as glycerin, causes refractive index matching between the scatterers and their surrounding media to reduce the scattering coefficient of biological tissues, which is often considered as one of the major mechanisms of TOC [23–25]. In fact, the optical properties of skins can be affected in many medical practices, such as the application of cosmetics or transdermal drug delivery [26,27].

MMI provide attractive tools for quantitative characterization of the skin conditions. However, in previous works using DRRs backscattering Mueller matrix imaging, we found that the TOC process in double-layer tissues can be complex. The DRRs system took 30 polarization component images in more than 3 min, which appeared too slow to capture the response of the TOC dynamics and could contain significant errors in the measurements.

In this paper, we use TOC by glycerin to generate dynamic variations of microstructure and penetration depths in single-layer and double-layer tissues. We established a fast-backscattering Mueller matrix imaging device based on division of focal plane polarimeters (DoFP) [11] with much faster speed and improved measurement accuracy. The device takes 17 s for each MMI measurement, compared with the backscattering MMI device based on DRRs, which takes 192 s [11]. In addition, to investigate the origin for the instability of the experimental results from the DRRs system, we examine in detail how the errors in the MM elements increase for longer intervals between successive measurements of polarization components or Stokes. By monitoring the TOC dynamics on double-layer and monolayer samples, we demonstrate that some MM elements and polarization parameters can characterize the double-layer features during TOC, which demonstrate that the fast-backscattering Mueller matrix imaging can be a powerful tool to probe and understand the dynamics variations in the microstructure of layered tissues.

## 2. Materials and Methods

### 2.1. Experimental Setup

The backscattering MMI system is based on DoFP which was successfully used in an upright transmission Mueller matrix microscope for fast MMI of tissue slides and cells [11]. In the system, as shown in Figure 1a, all cage system accessories were provided by RayCage(Zhenjiang) Photoelectric Technol-ogy Co., Ltd. Diffusing light from LED source (10 W, 633 nm, Cree) is collimated by a lens (L1, Daheng Optic, Beijing, China) and then modulated by the polarization states generator (PSG) which consists of a polarizer (P1, extinction ratio >1000:1, Daheng Optic, Beijing, China) and a quarter-wave plate (R1, LBTEK Optic, Changsha, China) fixed on a motorized rotation stage (PRM1/Z8, Thorlabs, Newton, NJ, USA). The backscattering photons from the sample are collected by a lens (L2, LBTEK Optic, China) and detected by polarization states analyzer (PSA), which includes 50:50 non-polarized beam splitter prism (NPBS, Thorlab, Newton, NJ, USA) and two DoFP polarimeters (DoFP 1 and DoFP2, Lucid Vision Labs, Canada) installed on the transmission and reflection ends of the NPBS, respectively. A quarter-wave plate (R1, LBTEK Optic, China) is installed between DoFP1 and the transmission end of the prism. The incident angle θ is about 15 degrees to avoid the surface reflection from the sample. The resolution

of the DoFP polarimeters is 2448 × 2048 pixels for 1.2 × 1.0 cm imaging area. Maximum errors among the matrix elements of standard samples are less than 1%.

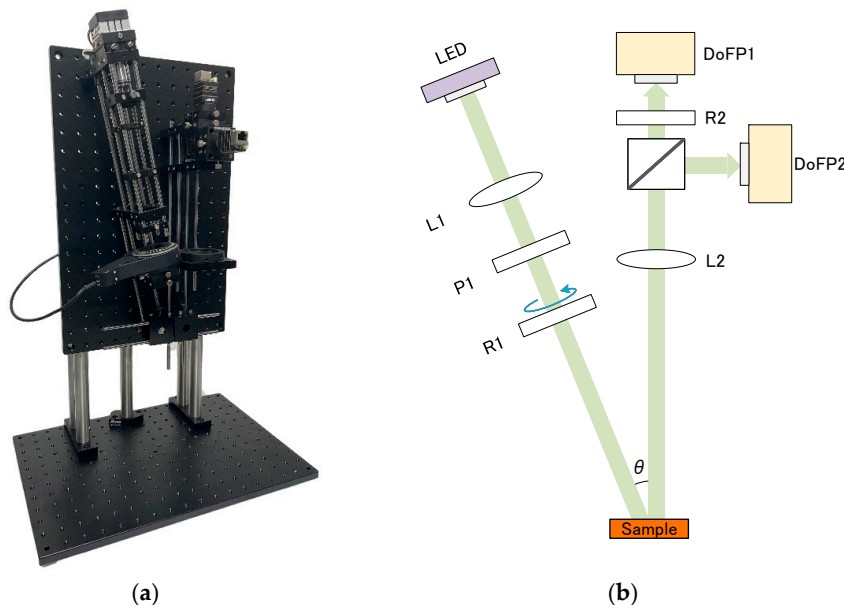

(**a**)                                                                (**b**)

**Figure 1.** Photograph (**a**) and configuration (**b**) of the fast-backscattering Mueller matrix system based on dual division of focal plane (DoFP) polarimeters.

### 2.2. Tissue Samples Preparation

As shown in Figure 2, we use human wrist inner skin (in vivo), single layer fresh tissue (ex vivo) and double-layer fresh tissue (ex vivo) as samples. The human skin (Figure 2a) was marked before experiments to ensure that the same position was measured each time. We applied 10% glycerin evenly on the mark area for 20 s, then cleaned the surface glycerin and took MMI on the same area once every 2 min over one hour.

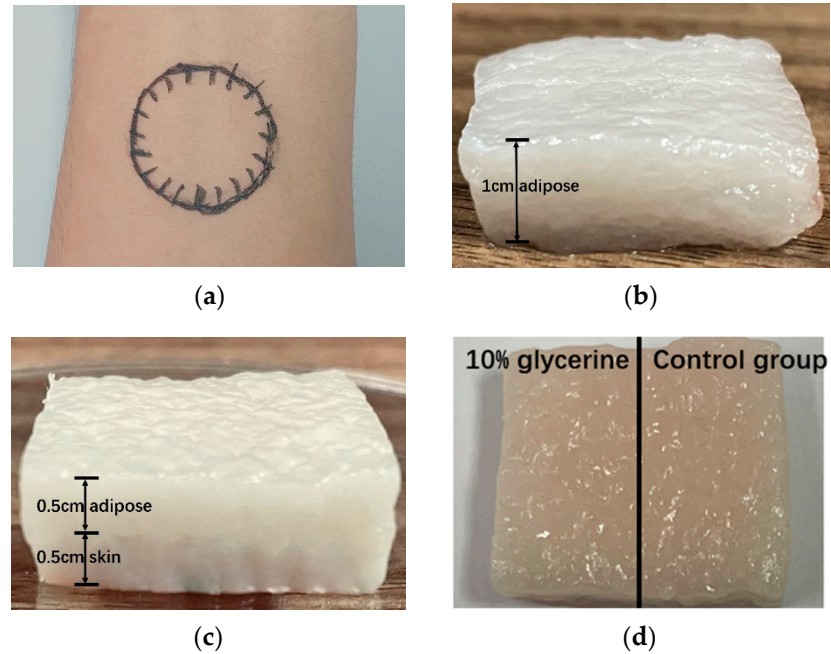

(**a**)                                                                (**b**)

(**c**)                                                                (**d**)

**Figure 2.** Tissue samples: (**a**) human inner wrist skin (anterior face), (**b**) single-layer tissue consisting of adipose layer, (**c**,**d**) double-layer tissue consisting of porcine skin and adipose layer.

The single layer and double-layer tissues used in this study were taken from the region of porcine hind legs, which were prepared 2 h after the porcine was butchered. The double-layer tissue (Figure 2c) consists of 0.5 cm thick porcine skin and 0.5 cm thick adipose layer. The single layer tissue (Figure 2b) was cut into a 1.5 × 1.5 cm cube of 1 cm thickness. The double-layer tissue (Figure 2c), consisting of 0.5 cm thick porcine skin and 0.5 cm thick adipose layer, was also cut into the same size as the single layer tissue, then was cut into two equal parts with a piece of plastic wrap between them (Figure 2d). During experiments, the double-layer sample were turned upside down with the adipose layer at the top and the porcine skin at the bottom. For TOC, 10% glycerin was applied evenly to the single layer tissue and one side of the bilayer tissue for 20 s. The other side of the double-layer tissue served as a control group, since the plastic wrap prevented lateral diffusion of the glycerin. For both single-layer and double-layer tissues, we cleaned the surface glycerin and measured them continuously for one hour.

*2.3. Polarization Parameters*

Our previous work has demonstrated that Mueller matrix and polarization parameters can be used as effective tools to study the microstructural variation due to interaction between clearing agents and biological tissues [28,29]. The elements of MM contain abundant polarization information of tissue. However, the relationship of a single MM element to a particular microstructure is unclear, so it is useful to extract polarization parameters from MM elements to quantitatively characterize the microstructure information of the sample [12,30].

In this work, we adopted Mueller matrix transformation (MMT) parameter *b* [31] to reflect the process of TOC, which relates to depolarization. Parameter *b* has been used in monitoring microstructural variations of fresh skeletal muscle tissues [32], probing layered structures by multi-color backscattering polarimetry [16]. In addition, we also applied MMT parameter *t*1 and Mueller matrix polar decomposition (MMPD) parameter *D* to the error analysis at different times of single MM measurements, which respectively represent anisotropy and diattenuation of the sample [33].

The equation expressions for the polarization parameter are shown in Equation (2).

$$
\begin{aligned}
b &= \frac{M22+M33}{2} \\
t1 &= \frac{(M22-M33)^2+(M23+M32)^2}{4} \\
D &= \sqrt{M12^2 + M13^2 + M14^2}
\end{aligned}
\tag{2}
$$

*2.4. Monte Carlo (MC) Simulation*

Due to the complexity of biological tissues, Monte Carlo (MC) simulation has been widely used in previous works to help us understand the microstructural changes and the optical properties of biological tissues [34,35]. In order to further understand the possible microstructural changes during TOC, Monte Carlo simulations based on the sphere-cylinder birefringence model (SCBM) are chosen to simulate the trajectory of photon scattering in tissues and the changes of their polarization states. The detailed parameters used in the MC simulations will be introduced in the following section.

## 3. Results and Discussions

*3.1. TOC Dynamics of Human Skin*

For backscattering MMI of human skin, only the diagonal elements of MM showed significant response during TOC. As shown in Figure 3, values of *M*22, *M*33 and *M*44 decrease for the first 35 min, then gradually increase and eventually level off with the experimental time. Such a clear turn during TOC indicates that the skin sample is a layered structure because the polarization features change sharply as photons penetrate to another layer of very different optical properties. The sensitive response of diagonal elements *M*22, *M*33 and *M*44 indicates that TOC affects mainly depolarization of human skin. Smaller values of the diagonal elements mean stronger depolarization. The difference between

$M$22 and $M$33 is small and the non-diagonal MM elements values are small, both of which indicate that human skin samples tend to be isotropic [32].

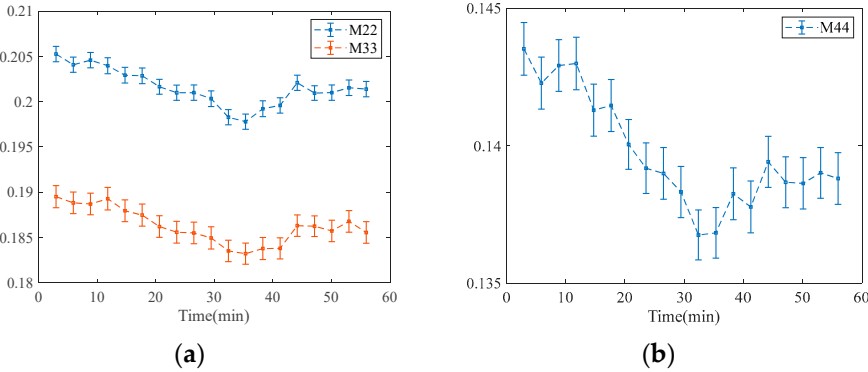

**Figure 3.** Average values and standard deviations of the MM diagonal elements of human skin: (**a**) $M$22, $M$33 and (**b**) $M$44. The horizontal axes represent the time after the application of 10% glycerin.

Since it is difficult to keep the wrist stationary for a long time, we prepared animal tissue samples for detailed studies in the following works.

### 3.2. Error Analysis at Different Time of Single MM Measurement

Before using a Mueller matrix device based on DoFP, we have taken MMI of human skin during TOC by dual rotating retarders (DRRs) Mueller matrix imaging, which recorded 30 polarization component images in about 3 min to derive the MM image. It was found that the experimental results were very unstable, unlike what was obtained with the DoFP-based device. Since the characteristic time scale of TOC for the human skin sample is over 20 min, we speculate that the instability of experimentally obtained MM is due to the high sensitivity of MMI to the duration component measurements.

To test the relationship between MMI accuracy and the duration between Stokes imaging, we varied the time between successive Stokes measurements, $DT_{Stokes}$, and compared the variation and error of the MM elements. The DoFP-based device takes a Stokes image in a snapshot but completes a MM measurement by rotating the quarter wave plate R1 to four different angles to generate four incident polarization states [11]. Using angle $\zeta_1$ data of first measurement, angle $\zeta_2$ data of the second measurement, angle $\zeta_3$ data of the third measurement and angle $\zeta_4$ data of the fourth measurement, we derive an MM image with four times longer intervals in Stokes imaging, $DT_{Stokes}$. In the same way, we can further extend the time interval by factors of 8, 12 and 16. The device takes 17 s minimum for each MM measurement, so we can extend the MM imaging time, $DT_{MM}$, to 68, 136, 204 and 272 s.

Figure 4 shows the average value of MM elements at different $DT_{MM}$ during TOC of the double-layer tissue. The MM elements demonstrate larger errors as $DT_{MM}$ increase: (1) the curves of the elements $M$13, $M$23 and $M$32 show significant differences as $DT_{MM}$ increases, while other non-diagonal elements remain almost constant. It indicates that the structural information related to elements $M$13, $M$23 and $M$32 will be affected, such as diattenuation and anisotropy; (2) the peak variations of diagonal elements $M$22, $M$33 and $M$44 indicate that the time error of probing deep information increases with the increase in $DT_{MM}$; and (3) the element $M$22 and $M$33 gradually fails to probe deeper information of double-layer tissue as $DT_{MM}$ increases. When the time is greater than 204 s, the characterization of deep information by $M$22 and $M$33 becomes less obvious, which can be a reasonable explanation for the instability of the experimental results from the DRRs system. It should be noted that, due to the difference in $DT_{MM}$, the starting points of the curves in the figure are different.

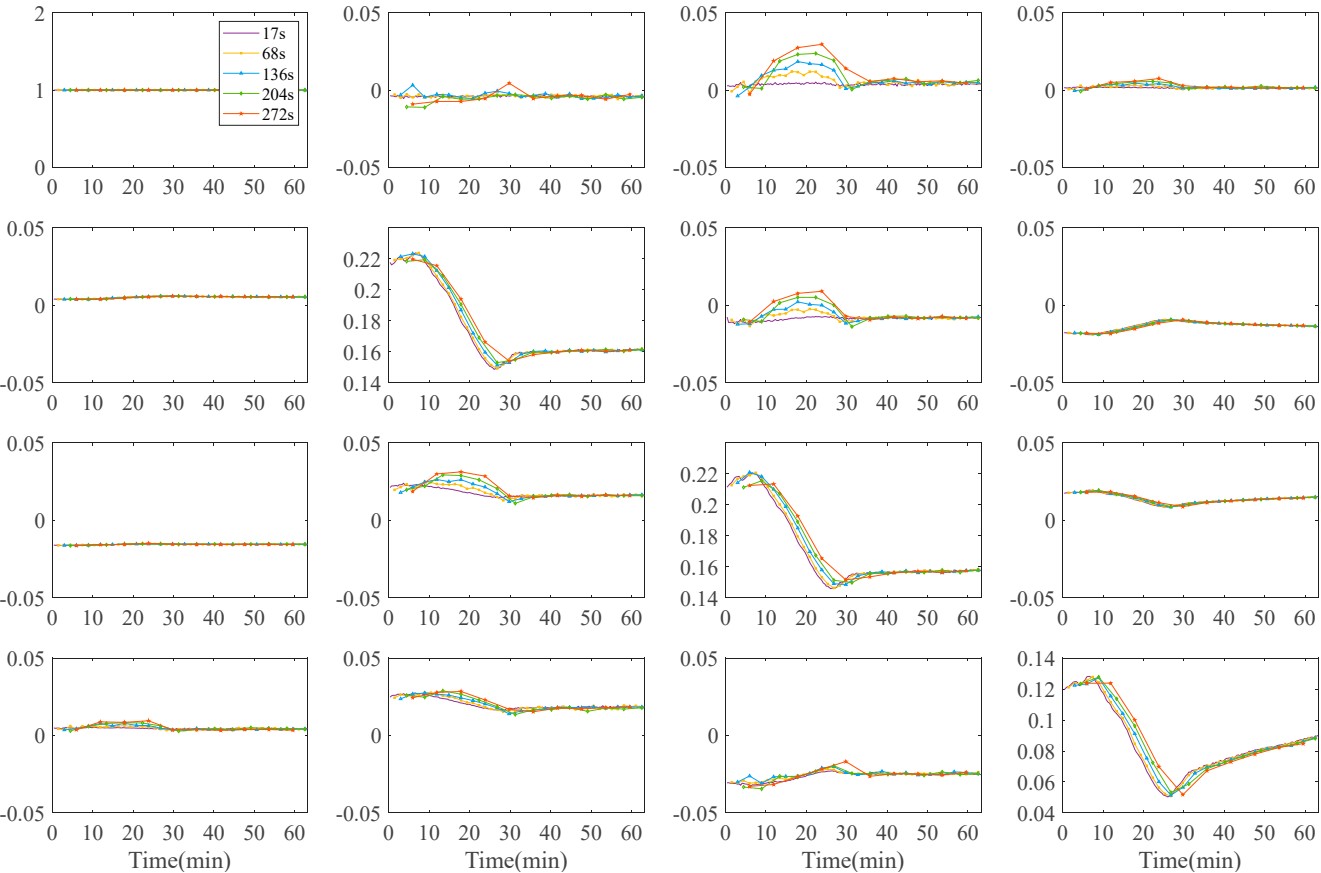

**Figure 4.** Average value of the MM elements for double-layer tissue at different times of single measurement: 17 s (the purple line), 68 s (the yellow line), 136 s (the blue line), 204 s (the green line) and 272 s (the red line). The horizontal axis represents the time after application of 10% glycerin to the double-layer tissue.

To further examine how the error of MM elements of MM vary with increasing $DT_{MM}$, we set the MM values at the minimum 17 s as the true value and those at other $DT_{MM}$ as the test value. As shown in Figure 5, we calculated factorized errors as the differences between the true and test values of the MM elements divided by the rate of change of the true value itself at different $DT_{MM}$. The equation expressions for the $MM_{Error}$ parameter are shown in Equation (3).

$$MM_{Error} = \frac{MM_{Measurement} - MM_{Ture}}{dMM_{Ture}/dt} \tag{3}$$

The result shows that such errors are always much bigger than the unit, which means that the MM element errors induced by slow acquisition of individual Stokes images, or $DT_{stokes} = DT_{MM}/4$, are much more sensitive than the variation of MM elements themselves. For reliable measurements of the dynamic processes using Mueller polarimetry, the Stokes imaging time has to be much faster than the time scales of the processes.

### 3.3. TOC Dynamics of Double-Layer Tissue
3.3.1. Mueller Matrix Elements of Double-Layer Tissue

Since TOC experiments on human skin showed layered phenomenon and it is difficult to take high quality long-time measurements on living skins, we prepared a double-layer tissue sample consisting of porcine skin and an adipose layer. Compared to measurements on human skins which took data at 2-min intervals, the double-layer tissue can be sampled

continuously at a minimum interval of 17 s for each measurement, so that more points can be obtained under the same experimental conditions.

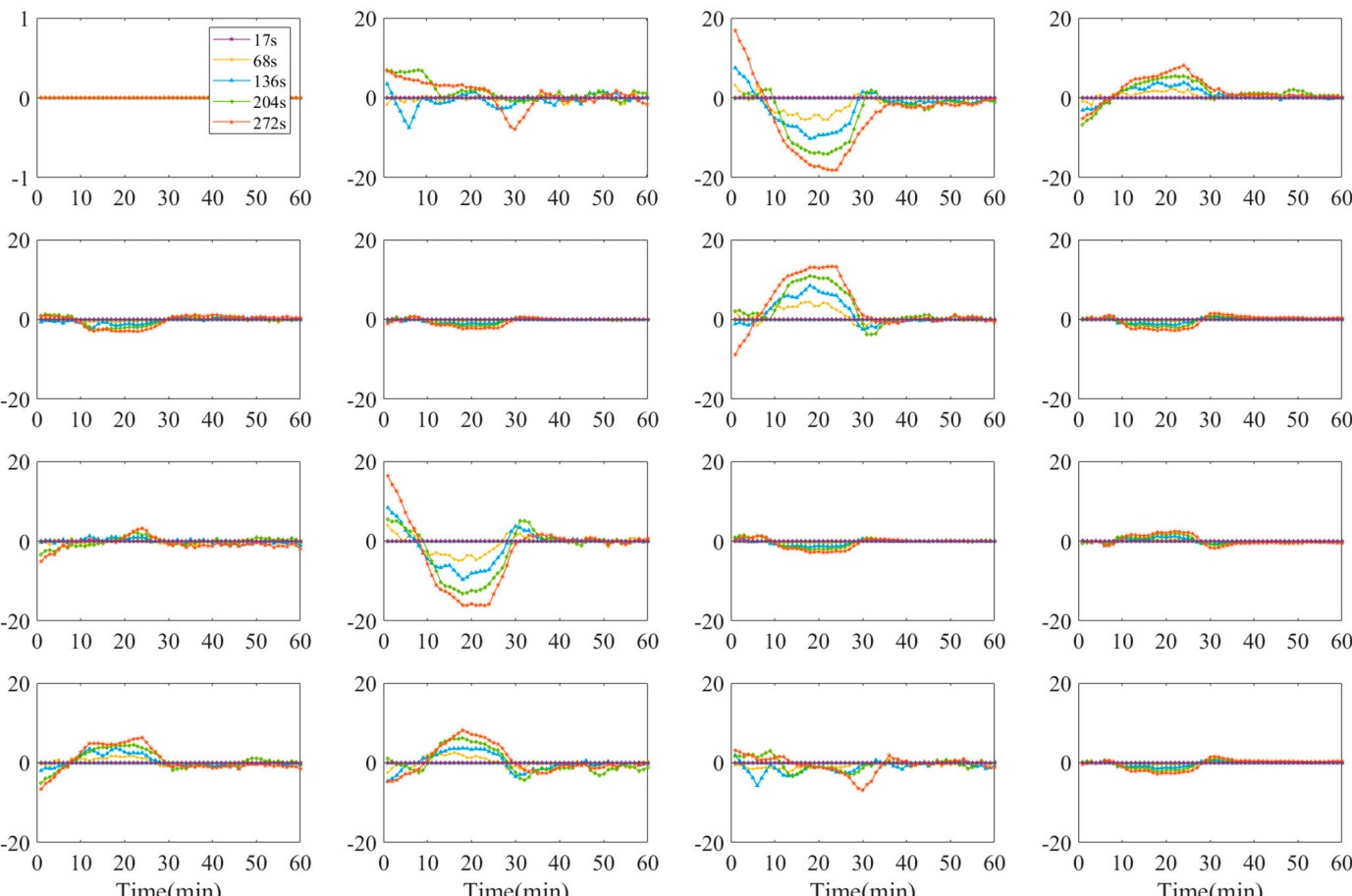

**Figure 5.** Relationship between the error of true value and test value and the rate of change of true value itself at different times of single measurement: 17 s (the purple line), 68 s (the yellow line), 136 s (the blue line), 204 s (the green line) and 272 s (the red line). The horizontal axis represents the time after application of 10% glycerin to the double-layer tissue and the vertical axis represents the value of the error value between the true and test values of the Mueller matrix element divided by the rate of change of the true value itself at different measurement times.

Figure 6 shows the MM images of double-layer tissue at different treatment in 8, 25 and 58 min, respectively. Figure 7 shows the mean values of MM elements of double-layer tissue with or without the application of 10% glycerin. The experimental results show that: (1) the values of all the three diagonal MM elements first decrease and then increase, which indicates depolarization power first increases and then decreases; (2) the small difference between $M22$ and $M33$ represents that the samples are mainly isotropic; (3) the values of the non-diagonal elements are small and barely change over the experimental time, which also indicates that depolarization plays a major role in the TOC process, whereas other factors such as birefringence and linear retardance make limited contributions; and (4) the characteristic trends of the MM diagonal elements during TOC also display a layered feature which is similar to the human skin.

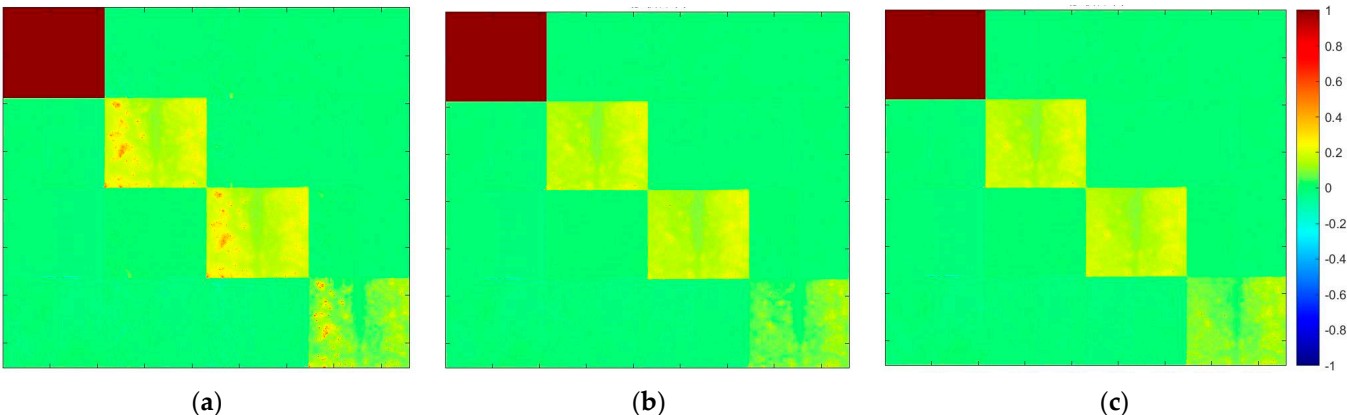

**Figure 6.** Images of Mueller matrices of double-layer tissue consisting of porcine skin and adipose layer in one-hour experimental time: (**a**) 8, (**b**) 25, (**c**) 58 min. Among the 16 matrix element images of the three MM images, the left half represents tissue with application of 10% glycerin and the right half represents the control group.

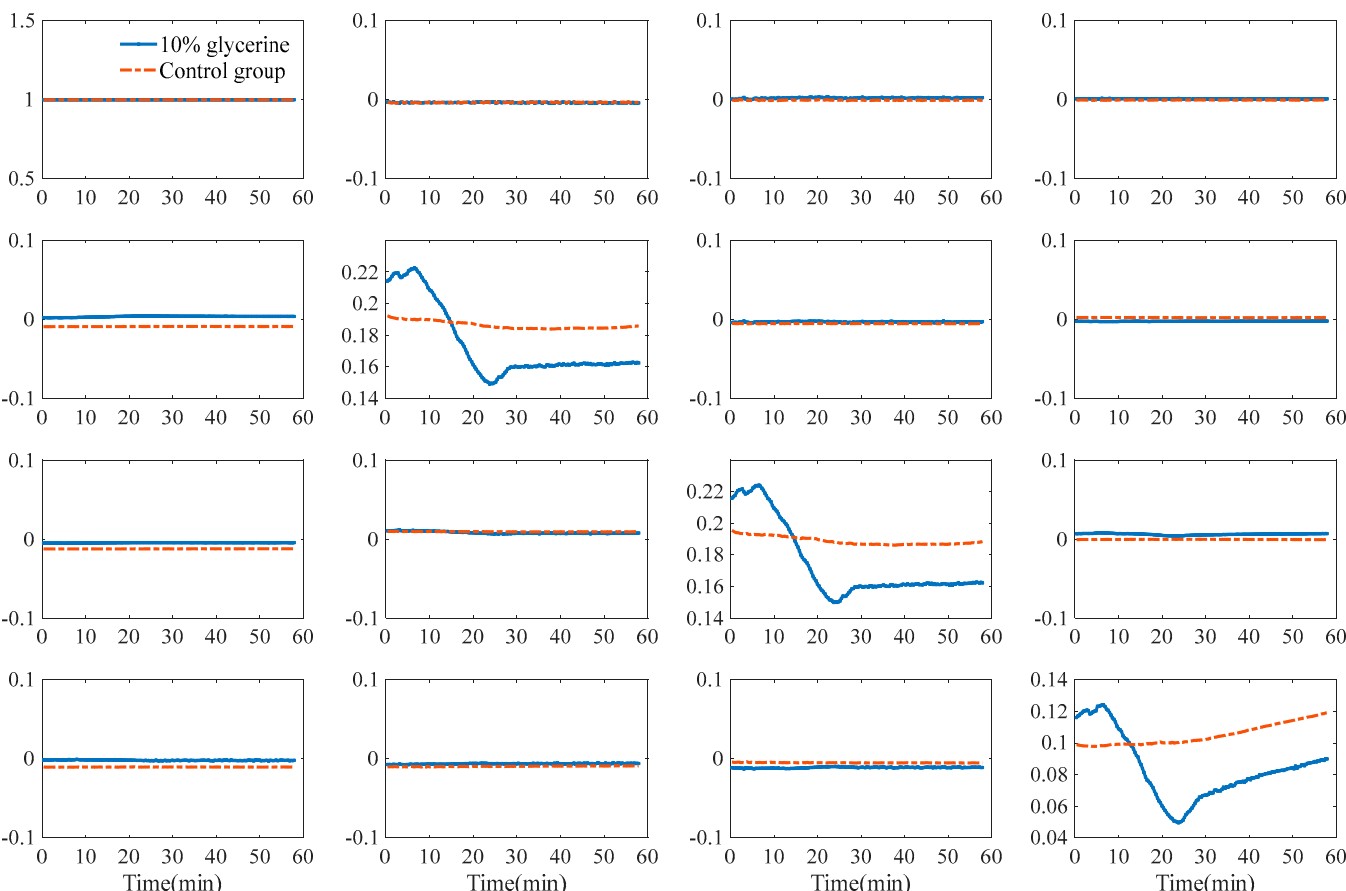

**Figure 7.** Average values elements of MM of double-layer tissue at different treatment: application of 10% glycerin (blue lines) and control group (red line). The horizontal axis represents the time after different treatment.

### 3.3.2. Polarization Parameters of Single-Layer and Double-Layer Tissues

We have found that MM elements can probe the double-layer tissue [16], but they may not clearly display the microstructure of the tissues. Polarization parameters with clearer physics meanings, such as *b*, are more suitable to probe layered tissue during TOC. Particularly, *b* is negatively correlated depolarization [31]. To further verify that the sharp

turn at approximately 25 min after TOC was a characteristic feature for layered structure, we measured TOC using single-layer adipose tissue of the same size and thickness as double-layer tissue with 10% glycerin. Figure 5 shows average values of parameter *b* for different samples.

As shown in Figure 8a, for double-layer tissue, the result shows that the structural variation can be divided into four stages. The 10 min after application of 10% glycerin can be considered as the first stage when *b* remains almost constant. The second stage is from 10 to 25 min when parameter *b* decreases significantly, which indicates that the depolarization power increases. From 25 to 30 min can be regarded as the third stage when parameter *b* increases. In addition, the fourth stage is from 30 to 60 min when parameter *b* remains nearly constant. By comparing the variation of parameters with single-layer and double-layer tissue as TOC proceeds, we found a clear peak at 25 min for double-layer tissue and none for single-layer tissue. The different trends in parameters for different samples indicate the ability of parameter *b* to probe deep information in the TOC process.

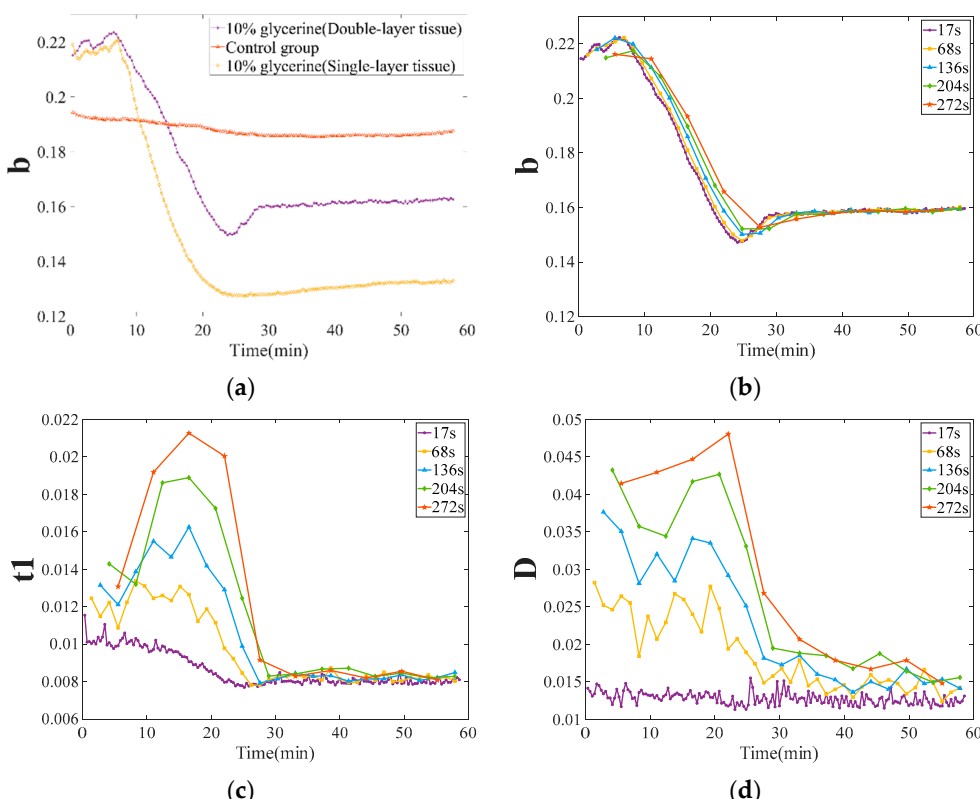

**Figure 8.** (**a**) Average values of MMT parameter *b* of different samples: double-layer tissue (porcine skin and adipose layer) with application of 10% glycerin (the blue line), double-layer tissue (porcine skin and adipose layer) without glycerin (the red line) as control and single-layer tissue (adipose layer) with application of 10% glycerin (the yellow line). Average value of parameters for double-layer tissue at different $DT_{MM}$: (**b**) MMT parameter *b*, (**c**) MMT parameter *t*1 and (**d**) MMPD parameter *D*. The horizontal axis represents the time after application of 10% glycerin to the double-layer tissue.

In this section, we also compared the variations of polarization parameters at different $DT_{MM}$. Figure 8b–d shows the average values of MM elements at different $DT_{MM}$ during TOC of the double-layer tissue: (1) the variation of parameter *b* shows that the position of sharp turn at approximately 25 min is gradually delayed as $DT_{MM}$ increases, indicating the time error increase by degrees; (2) the above analysis proves that the sample tends to be isotropic, indicating both anisotropy (MMT parameter *t*1) and diattenuation (MMPD parameter *D*) are in a small range, which are shown in Figure 8c,d. when $DT_{MM}$ = 17 s. However, the illusion of increase appears in both parameters *D* and *t*1 with the increase in

$DT_{MM}$, resulting in the wrong experimental phenomenon. The result shows fast MMI is necessary in subsequent measurements.

### 3.3.3. Monte Carlo (MC) Simulations

In order to better understand the changes in microstructure of the double-layer samples during TOC, we simulate the backward Mueller matrices of the scatterer model (SSM) and the sphere-cylinder birefringence model (SCBM) with Monte Carlo simulations corresponding to the samples of different layers, respectively. The simulation results are compared with experimental results to verify their validity.

For the double-layer samples constructed from porcine adipose and skin used in this paper, we constructed two different optical models corresponding to the two types of structures. Porcine adipose is a typical optically isotropic sample, so we used the spherical scatterer model (SSM), which is an isotropic scatterer, to simulate it with the following parameters: the diameter of the spherical scatterer changes from 0.2 to 0.25 μm with an interval of 0.01 μm, the scattering coefficient is 140 cm$^{-1}$, the refractive index is 1.49 [36] and the refractive index of the medium is 1.35. The porcine skin is rich in collagen fibers, so we used the sphere-cylinder birefringence model (SCBM) for its simulation. The diameters of the scatterers are 0.2 μm for the spheres and 1.5 μm for the cylinders. The scattering coefficients for spherical and cylindrical scatterers are 10 and 190 cm$^{-1}$, respectively [28]. The azimuth angle of the cylinder scatterer is 0° and the rise and fall varies gradually from 40° at −2° intervals to 34° with the course of the TOC. The zenith angle of the cylinder scatterer is 90°, and the rise and fall is 10°. The refractive indices of both spherical and cylindrical scatterers are 1.43, the birefringence value is $3.0 \times 10^{-5}$ and the refractive index of the medium is 1.35. For both models, the incident light wavelength is 633 nm, the thickness is 0.5 cm and the number of photons for a simulation is $10^7$. We then combine the experimental and simulated data together to analyze and discuss the possible mechanism of tissue permeability.

The refractive index matching between the scatterer and the surrounding interstitial in biological tissues is often considered as one of the major mechanisms of TOC [37]. The high permeability of glycerin solutions leads to an increase in the refractive index of the interstitial mass of biological tissues, resulting in a gradual decrease in the refractive index difference between the scatterer and the interstitial mass, thus reducing scattering from the tissue. For this reason, we gradually change the refractive index of the interstitial mass from 1.35 with 0.01 as an interval to 1.40 to simulate the refractive index matching process.

For the double-layer samples, we discuss them in layers. The first layer consists of porcine adipose cells. There have been many experiments and theories linking the optical properties of biological tissue scatterers to their particle size [38]. Combining the experimental data, it can be clearly observed that when the photons mainly pass through the first layer of samples, before the inflection point appears, $M22 > M44$ [39], which means that the light we observe is mainly scattered by particles that are small compared to the measured wavelength, so the particle size of the spherical scatterers in the first layer of the SSM is smaller, with a diameter of 0.2 μm. Considering that the second layer of porcine skin samples is rich in fibrous structures, such as collagen fibers, a SCBM is used. For the effect of tissue permeability on the double-layer samples, we change the refractive index of the two samples media separately during the simulation as a way to achieve refractive index matching.

As can be seen in Figure 9, for the *b* parameter, the simulation process is similar to that of TOC in the experiment. The line preservation bias gradually decreases when tissue permeabilization is performed for isotropic tissues. When TOC is performed for anisotropic media containing coarse fibers, the refractive index matching is accompanied by the gradual ordering of the cylinder scatterers due to water loss, and their backscattering line preservation gradually increases. When the refractive index matching ends, the b-value tends to stabilize.

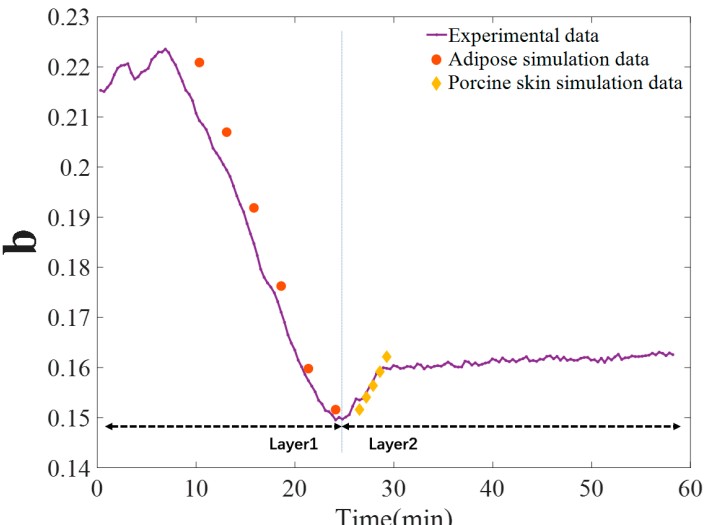

**Figure 9.** Average values of MMT parameter *b* and the result of MC simulations for double-layer tissue. Layer1: refractive index of the interstitial mass from 1.35 with 0.01 as an interval to 1.40. Layer2: refractive index of the interstitial mass from 1.35 with 0.01 as an interval to 1.39.

## 4. Conclusions

In this paper, we set up a fast-backscattering Mueller matrix imaging (MMI) device based on DoFP. The device takes 17 s for each MMI measurement which is much faster than 192 s for the DRRs-based device. We used this DoFP-based device to examine tissue optical clearing (TOC) on human skins and found that the diagonal elements of MM respond sensitively to TOC and display a characteristic sharp turn as TOC proceeds, which is an indication that photons penetrate another layer of a different optical property. For further studies of the TOC process of skin, we established single-layer and double-layer tissue samples. We investigated the reasons for the instability of MMI in previous studies of TOC using the slow DRRs-based device. By varying the time interval between successive Stokes images, from which the MM images are calculated, and examining the corresponding errors in the MM elements, we proved that such time interval has significant effects on the accuracy of MM elements during TOC process. For reliable measurements of the dynamic processes using Mueller polarimetry, the Stokes images have to be captured in much shorter intervals than the time scales of the dynamic process. We also examined the characteristic behaviors of Mueller matrix elements and polarization parameters with clear physics meanings, which demonstrates that the fast backscatter Mueller matrix imaging can be a powerful tool to probe dynamics of interaction between clearing agents and layered tissues. MC simulations based on SCBM proved that refractive index matching can explain the TOC process of the double-layer tissue using 10% glycerin as a clearing agent. The result shows that the samples are nearly isotropic, diagonal elements and parameter b, all of which are related to depolarization, respond sensitively to the layered structure during TOC.

**Author Contributions:** Conceptualization, H.M.; data curation, T.B., C.S., Q.Z., Y.Z., Y.S. and Y.W.; methodology, T.H., Y.Z. and Q.Z.; supervision, H.M.; writing—original draft, T.B. and C.S.; writing—review and editing, T.B., C.S. and H.M. All authors have read and agreed to the published version of the manuscript.

**Funding:** This research was funded by the National Natural Science Foundation of China (Grants: 61527826 and 11974206) and the Shenzhen Bureau of Science and Innovation (Grants: JCYJ20170412170814624 and JCYJ20160818143050110).

**Institutional Review Board Statement:** The study was conducted according to the guidelines of the Declaration of Helsinki, and approved by Ethics Committee of Tsinghua University (Ethics Issue No. 9, 2022).

**Informed Consent Statement:** Informed consent was obtained from all subjects involved in the study. Written informed consent has been obtained from the patient(s) to publish this paper.

**Data Availability Statement:** The data presented in this study are available on request from the corresponding author. The data are not publicly available due to privacy.

**Conflicts of Interest:** The authors declare no conflict of interest.

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
