# Peer review of "Probing Dynamic Variation of Layered Microstructure Using Backscattering Polarization Imaging"

_photonics, doi:10.3390/photonics9030153_

Round 1

Reviewer 1 Report

The methods and approaches that you used in your article are quite interesting. After analyzing the list of sources used, I can recommend a series of articles by scientists from Chernivtsi National University under the guidance of Professor Oleksandr Ushenka, who also have highly rated articles on this topic.

Author Response

Thank you very much for your valuable comments, I have gained a lot from your comments.

The attached document is a response to your comments.

Reviewer 2 Report

The article aims to establish an improved imaging device allowing the measurement of the optical 
clearing of human skin samples.

The article is interesting but I report some important points to clarify, please.

Specific comments :
1-lines 29-34: This first paragraph is too short and lacks some definition and details to clarify key elements necessary to understand the rest of the study and its importance. 
Please explain what is polarization imaging, present the biological tissue you are working on and its characteristics (double-layer, ...), give a simple definition of the Muller matrix.

2-lines 35-39: Please describe the tissue optical clearing method you used, and why did you choose it rather than another one.

3-2.2 Tissue sample preparation, line 87-104: Please, this part needs to be clarified. 
What tissue optical clearing method did you use? Precise the protocol associated.
The adipose layer (Figure 2. b), is not cited in the text, no information is given about its origin.
For the human skin, you applied 10% of glycerin, did you also do some control without like for the porcine sample? If not, why?
What is a qualitative filter paper? And why did you use it?

4-lines 116-117: Please, I need clarifications about the usage of Muller Matrix to study the process of tissue optical clearing. As you use your process on an optically cleared sample (I refer to the porcine skin sample), without a control (not optically cleared porcine skin sample) nor a live acquisition during the tissue optical clearing process (not impossible I presume but really difficult I assume), how can you be sure that your Muller Matrix reflect tissue optical clearing effect?

5-3.Results and discussions: Please add images of what you visualized by the Mueller matrix system based on the dual division of focal plane imaging technique to give illustrations of the possible results.

6-3. Results and discussions: Please precise in figures caption the name of the sample as presented in the Material and Methods. For instant in the Figure 7 caption, lines 237-238, "double-layer tissue (porcine skin) with 10% of glycerine, double-layer tissue (porcine skin) without glycerine as Control, ..."

7-lines 138-139: Please I need clarifications. Did you also optically clear human skin?

Author Response

(The authors gave the same response as above.)

Reviewer 3 Report

Review

Article ID photonics-1560471

Title:  Probing tissue optical clearing dynamics of layered samples

First author: Tongjun Bu ; Journal: Photonics (MDPI)

Authors show results obtained with backscattering Mueller matrix system allowing faster measurements and therefore being more appropriate for in vivo measurements. Authors developped an interesting method allowing to test the sensitivity of the measurement to optical clearing on both types of skin samples (single and two-layered samples) using glycerin as an optical clearing agent.

Introduction should be completed in order to help the reader who doesn’t know about polarization what the goal of the study is (except for the more rapid emasurement which is well described) in terms of double layered-tissue sample.

The description of tissue optical clearing is a little naive (lines 35-39): « makes the tissue more transparent » « homogenize small particles ».When talking about « more transparent », do you mean less scattering? When talking about « small particles », are you talking about free water that is replaced by the optical clearing agent which induces refractive index matching? Please refer to articles describing the phenomenon of tissue optical clearing in order to provide the reader with thourough information on this e.g. Zaytsev et al., J Biophotonics, 2022, 15 (1), DOI: 10.1002/jbio.202100202

Please provide information about patients’s informed consent and ethical approval of experiments on human beings as well as an ethical committee approval for experiments on animals tissues: on which types of animals (pig race) were adipose tissues removed (from what part of the pig ?) and were animals used only for this purpose or also for additional purposesas well?

Please provide more information on glycerin as an optical clearing agent and why authors chose such an optical clearing agent and not another one? What about its toxicity and clearing efficiency compared to other types of clearing agents?

Please mention why you did not perform several measurements on the same person (e.g. two skin sites on the right wrist and 2 skin sites on the left wrist) in order to measure standard deviation of the measure ?

Line 88-92 : this paragraph is difficult to understand so the reader may not undestand clearly the method. I propose that authors mention « ex vivo » and « in vivo » experiments so it might be easier to understand the difference between the experiment performed on human skin on one hand (which is considered as a double-layered sample), and on porcine tissues on the other hand (on a single layered and on a double layered-sample for the ex vivo experiment).

In this same paragraph, do authors mean that they applied glycerin as an optical clearing agent only for 20 s on each sample beofre making measurement?

Please provide the reference of the « qualitative filter paper » and describe what it is made of.

It seems like in vivo measurement is performed on human INNER wrist (anterior face). Please add the information.

As a general comment, the English level should be improved (I listed just few mistakes below but there are many more).

Line 31 : biological tissueS, « s » is missing.

Li,es 31-33 : the sentence doens’t make sense, should the word « and » (line 32) be discarded ?

Line 88 : « as shown in Figure 2, We use… », the uppercase « w » should be replaced by a lowercase « w ».

Line 126 : « in previous workS », « s » is missing.

Author Response

(The authors gave the same response as above.)

Round 2

Reviewer 2 Report

I thank the authors for their clarifications and the text modifications that made the manuscript easier to read and understand.

The manuscript is almost suitable for publication, there still is only one important point to clarify :

The last point concerns the "Institutional Review Board Statement and approval number for studies" that is not presented while the study involves the usage of humans and animal samples.

If added, I recommend the manuscript for publication.